# Adolescents and Resilience: Factors Contributing to Health-Related Quality of Life during the COVID-19 Pandemic

**DOI:** 10.3390/ijerph19063157

**Published:** 2022-03-08

**Authors:** Miri Tal-Saban, Shahar Zaguri-Vittenberg

**Affiliations:** School of Occupational Therapy, Hebrew University of Jerusalem, Jerusalem 91240, Israel; shahar.wittenberg@mail.huji.ac.il

**Keywords:** adolescents, health, quality of life, sense of coherence, hope, coronavirus

## Abstract

This study aimed to examine health-related quality of life of adolescents before and during the COVID-19 outbreak, and its relationship to resilience embodied in hope and a sense of coherence. Typically developed adolescents between the ages of 13 to 18 participated in the study; 84 were recruited before the pandemic outbreak and 64 in March to April 2020 during the worldwide outbreak. The participants completed the Pediatric Quality of Life Inventory, Sense of Coherence Scale, and Children’s Hope Scale. During the outbreak, adolescents reported higher physical health-related quality of life scores (F(1146) = 3.710, *p* < 0.05, η² = 0.027) and lower school health-related quality of life scores (F(1146) = 5.748, *p* < 0.05, η^2^ = 0.028), compared to adolescents during the pre-outbreak period. Furthermore, adolescents during the outbreak reported a significantly (*p* < 0.05) higher sense of coherence but no difference in levels of hope. Finally, the results of multiple linear regression indicated that resilience factors (hope and sense of coherence) contributed to the prediction of health-related quality of life, independent of socio-demographic variables. Hope and a sense of coherence were both found to positively impact one core aspect of health, which highlights the importance of addressing resilience factors in educational and therapeutic settings for adolescents thus that they are better able to adapt to stressful events such as pandemics.

## 1. Introduction

The latest coronavirus (COVID-19) leads to or causes an infectious disease. Coronavirus outbreaks and declines are rapid, causing global morbidity and mortality to rise. As a result, the international community is coping with the pandemic’s impact on almost all aspects of daily life, including social distancing, isolation, and changes in life routines [1]. 

Health-related quality of life (HRQoL) is a subjective dynamic appraisal of one’s position in life in the context of internal views (values, interests, goals, concerns, and expectations) and external factors (e.g., culture). It is influenced by physical health, psychological state, level of independence, social connections, and environmental settings [2]. Quality of life (QoL) is an essential outcome measure for both general and mental health. The research literature recognizes the vital importance of QoL measures in addressing adolescents’ health [3,4,5]. Stressful events such as war or pandemics can cause QoL disruptions. Indeed, worldwide studies described QoL deterioration during the COVID-19 outbreak, with greater distress levels and reduced QoL among women and individuals with poor health [6]. 

Adolescence is a transitional stage between the ages of 10–19 years, beginning with the onset of physiological maturation and ending with a developed sense of self, identity, and behavior [7]. Due to rapid physical changes, adolescents may experience an imbalance in physiological processes that can manifest as impaired personal well-being and HRQoL [8]. The COVID-19 pandemic has caused unprecedented changes in the lives of millions of adolescents worldwide [9], harming their engagement in positive QoL predictors such as physical activity [10] or socialization with family members and peers [11]. For this reason, adolescents are particularly vulnerable to secondary adverse impacts on mental health in the ongoing COVID-19 situation [12,13]. A review by Nobari et al. [14] described inconsistency in research findings regarding HRQoL among adolescents. While most studies reported reduced HRQoL among adolescents during the pandemic compared to the period before it, others found no change in HRQoL during the pandemic [14].

“Resilience” is a construct associated with the ability to cope with challenges raised by stressful difficulties or to adapt despite experiencing difficulty in circumstances [15]. Stress-coping theories suggest that different internal and external personal factors can protect from stress-induced adversities and aid resilience [16]. This study focused on two such factors: a sense of coherence (SOC) [17] and hope [18].

“Hope” is a motivational cognitive structure that represents one’s perceived capability to invoke clear ways to fulfill personal goals, which may involve encountering stressors while pursuing them [18]. This cognitive array is composed of “pathway thinking,” the ability to use different means in efficient ways, and “agency,” the perception of one’s ability to succeed in the initiative and persevere in action until the desired goal is achieved [19]. Hence, individuals with high levels of hope can challenge themselves with long-term goals, are likely to find alternative pathways to reach goals when facing difficulties, and tend to view the stressor as challengeable [20]. 

Antonovsky [17], in his salutogenic model, explained how people manage to stay healthy despite stressful life events. Antonovsky claimed that an individuals’ life orientation has an impact on his health and influence by his sense of coherence (SOC). SOC is a concept that describes the current degree to which a person perceives events in the world as comprehensible, manageable, and meaningful. Comprehensibility is the ability to recognize that life is understandable, that things happen for a reason. Manageability is the sense that things can be managed and that there is flexibility in choosing strategies. Finally, meaningfulness is an understanding that demands are actually challenges that are worth investing in [17,21]. While hope conceptualizes individuals’ anticipation for their future, SOC reflects the developmental outcomes of past experiences and an all-encompassing life orientation that influences health [17]. Therefore, individuals with a strong SOC are likely to perceive stressful events as structured (coherent) rather than chaotic, will be confident of meeting demands, and tend to demonstrate determination and varied coping strategies when facing difficulties [22]. 

Resilience factors, life situations, and HRQoL have reciprocal relationships (See Figure 1). Both hope and SOC may be changed and enhanced over time as a result of life situations such as culture, social resources, and past stress exposures [17,19]. At the same time, hopeful thinking and SOC contribute to the attainment of good health; thus, both structures correlate with functional abilities, well-being [19], and measures of psychological health across an individual’s lifespan [18]. Unsurprisingly, both structures protect against reduced HRQoL [23,24]. 

### Aims

The aims of our study were: (1) to evaluate levels of HRQoL, hope, and SOC among adolescents during the COVID-19 pandemic outbreak, compared to before the pandemic; and (2) to evaluate the relationship and impact of resilience factors (SOC, hope) on adolescents’ HRQoL before and during the COVID-19 outbreak.

## 2. Materials and Methods

### 2.1. Participants

A total of 148 adolescents were recruited from all over Israel in 2 waves: (a) 84 in early 2019, before the COVID-19 outbreak (Age: Mean = 15.15; SD = 1.59; 51.2% girls); and (b) 64 in April–May 2020, during the outbreak (Age: Mean = 14.77; SD = 1.93; 50% girls). The sample size was calculated using G-power software [25] and was based on expected effects given previous studies [10]. With an expectation of 90% power and type-I error of 0.05, it was estimated that a sample of 146 participants was needed for moderate effect sizes. The inclusion criteria were aged 13–18 years, Hebrew-speaking, normal intelligence (assumed by attending a general educational framework), and without a major health condition (e.g., cerebral palsy, epilepsy, psychiatric diagnosis, head injury, etc.) based on parental reports.

The sample was matched by gender, age, and parents’ marital status (Table 1). However, the groups varied in parental education level. Thus, follow-up univariate analysis of variance (ANOVA) was performed to examine the effect of parents’ education on the study variables (HRQoL, SOC, and hope); no significant differences were revealed regarding mother’s education (PedsQL Total: F(2143) = 1.205, *p* = 0.3; SOC: F(2145) = 0.174, *p* = 0.84; hope: F(2145) = 1.00, *p* = 0.91) or father’s education (PedsQL Total: F(2143) = 0.197, *p* = 0.821; SOC: F(2145) = 0.876, *p* = 0.42; hope: (F(2145) = 0.377, *p* = 0.69). Based on these findings, parental education level was not controlled in the data analyses.

### 2.2. Measurements

Demographic questionnaires provided information on the adolescents (age, gender) and their parents (education, marital status). 

The Pediatric Quality of Life Inventory Version 4.0 (PedsQL) [26] is a self-report generic measure of HRQoL among adolescents aged 13–18. It consists of 23 items rating the frequency of experiencing difficulty in performing an action on a 5-point scale ranging from 0 (“never”) to 4 (“almost always”). The items were then reversed, summed, and transformed to a 0–100 subscale to create a health profile in 4 domains: physical, emotional, social, and school-related. Additionally, a psychosocial health summary score is calculated using the last 3 subscales; the total score is calculated based on the 4 subscales. Higher scores indicate higher HRQoL and fewer difficulties. The PedsQL has been shown to be a reliable and valid questionnaire [26]. The PedsQL 4.0 Hebrew version, used in our study, was linguistically validated and demonstrated high internal consistency (0.91) for adolescents [26].

The Sense of Coherence Scale-Short Version [17] is a self-report measuring a sense of coherence, using 14 items. Adolescents were asked to rate the items on a 7-point scale ranging from 1 (“never”) to 7 (“always”). The total score was the average of all items; higher scores reflect a higher sense of coherence. Evaluation of the scale’s Hebrew version showed internal consistency of 0.77 [27]. 

The Children’s Hope Scale [28] measures indicators of hope based on a self-report of 6 items that represent goal-directed thinking (agency) and process-related thinking (pathways). Adolescents rate their frequency of thinking or doing things on a 6-point Likert scale ranging from 1 (“never”) to 6 (“always”). Higher scores on the total mean score indicate higher levels of hopeful thinking. The scale’s Hebrew version showed good internal consistency ranging from 0.72 to 0.89 and test–retest reliability of r = 0.71–0.73 over one month [29].

### 2.3. Procedure

After ethical approval from the Hebrew University institutional review board (No. 14062019), a convenience sampling of participants was recruited via social media. Researchers contacted families of adolescents in Israel who expressed an interest in participating in our study. Both study groups were recruited in the same manner (as described above) and received the same explanation about the purpose of the study: exploring resilience and quality of life among typically developing adolescents. Parents and adolescents provided informed consent and completed demographic questionnaires; the adolescents completed the questionnaires using electronic forms. 

### 2.4. Data Analysis

Statistical analyses were performed using IBM SPSS (Version 25; IBM Corp., Armonk, NY, USA). The chi-square, Fisher’s exact tests, and t-tests were conducted to compare the demographic characteristics of the two groups. Multiple analysis of variance (MANOVA) examined group differences on the PedsQL, and t-tests examined group differences on the SOC and Hope Scales. Correlations between the measures were examined using Pearson correlations, and Fisher’s Z transformation compared correlation values between groups. Finally, to evaluate the influence of socio-demographics, SOC, hope, and COVID-19 (as a group) on the PedsQL total score, a multiple linear regression analysis was performed. The significance level for obtaining the research hypothesis was *p* < 0.05.

## 3. Results

The results of the MANOVA analysis of the PedsQL scales yielded significant group differences (*F*(4143) = 3.254, *p* = 0.014, η² = 0.086). In order to examine the source of this difference, the data from each scale were subjected to univariate analyses. This post-examination revealed significantly higher physical HRQoL and lower school HRQoL among adolescents during the COVID-19 outbreak compared to before it. Additional analysis comparing the psychological HRQoL scale and the PedsQL total score revealed no significant differences between groups. However, the mean scores of these scales were lower during the outbreak compared to before it (Table 2). 

There was a significant group effect on the SOC scale (t_(100.577)_ = 2.032, *p* = 0.04, *d* = 0.35); adolescents pre-COVID-19 reported lower SOC (M = 4.27, SD = 0.57), compared to adolescents during COVID-19 (M = 4.52, SD = 0.89). However, no significant difference (t_(146)_ = −0.27, *p* = 0.79, *d* = 0.05) emerged between pre-COVID-19 and COVID-19 groups on the Hope Scale (M = 4.55, SD = 0.92; and M = 4.51, SD = 0.74, respectively).

The correlation analysis indicated significant, moderate correlations between overall scores and subscales of the PedsQL, SOC, and Hope Scales, in both groups, except for a non-significant correlation between physical subscales of the PedsQL and SOC scale.

Comparing pre-COVID-19 and COVID-19 correlations revealed no significant difference, except the values between physical HRQoL and SOC (Z = −2.807, *p* = 0.00). This was significantly stronger in the COVID-19 group (Table 3).

Regression analysis, explaining the PedsQL total score with demographic characteristics, resilience factors, and study group (before or during COVID-19) as predictors, revealed significant results (*F*(7126) = 12.175, *p* = 0.00; R^2^ = 0.403). Meanwhile, only the SOC and Hope Scales contributed significantly (*p* < 0.05) to variability (Table 4).

## 4. Discussion

Considering the detrimental effects of the COVID-19 pandemic on HRQoL generally, and among adolescents in particular [14], as well as the possible protective effects of hope and SOC against HRQoL disruptions [23,30], the purpose of this study was to examine adolescents’ HRQoL and resilience factors (hope and SOC) during the first COVID-19 lockdown, compared to before COVID-19. No previous studies addressed these resilience factors during the COVID-19 outbreak, and their relationship to HRQoL were found. Our study endeavored to take a closer look at the relationship between these variables among adolescents before and during the COVID-19 pandemic. The results indicated overall significant differences between the study groups regarding HRQoL. However, upon closer examination of the total score in the HRQoL questionnaire, no significant differences were found between the study groups. Yet lower scores were recorded during the outbreak, indicating lower HRQoL during that time. While the impact of COVID-19 on mental well-being among adolescents (e.g., levels of stress, anxiety, depression, etc.) has been widely studied [12], there were only limited numbers of studies addressing the construct of HRQoL during the COVID-19 pandemic and their results were inconsistent. Some authors [31,32] did not identify decreased HRQoL among adolescents during the COVID-19 outbreak. However, other studies described reduced psychological and overall HRQoL [9,33]. Our results may be explained by the differences between Israel, where our study took place, and other countries where those studies were conducted [31,32]. Compared to other OECD countries, at the time of data collection, Israel was experiencing lower morbidity and very low mortality rates. Additionally, the Israeli government’s coping strategy was to order multiple lockdowns quickly, thus controlling the pandemic [1]. It is interesting to note that two large-scale studies conducted in Germany during different stages of the pandemic, using the same HRQoL measure (KIDSCREEN-10), showed inconsecutive results. Ravens-Sieberer et al. [33], who collected data in June 2020, reported disruption of HRQoL during the outbreak, while Koenig et al. [32], who collected data on August 2020, did not find differences in HRQoL during the outbreak. In terms of resilience during exposure to traumatic events, this inconsistency can be related to previous work documenting the wide impact of environmental and contextual factors on building resilience [16,34]. The inconsistencies in data collected worldwide on the impact of COVID-19 on adolescents’ HRQoL demand further investigation.

The results regarding HRQoL and the inconsistencies with previous studies should be interpreted in light of the specific HRQoL measure used in the current study. The PedsQL, is a generic measure that addresses health profiles across four domains and is not specifically tailored to stressful events such as the COVID-19 pandemic. It has been noted before that specific HRQoL instruments (e.g., with elements appropriate to this particular situation) are more sensitive in detecting deviations occurring over time [5].

Furthermore, the inconsistency with studies that found a reduction in HRQoL during the pandemic is likely due to the psychometric properties of the PedsQL. According to Feeny et al. [4], self-reported HRQoL measures are affected by age, because language comprehension and symbolic reasoning improve as one gets older. Consequently, it is possible that the younger children in this study did not comprehend the items the same as the older ones. It may also be true of the SOC and hope scales. They [4] suggested using proxy parent reports in pediatrics to overcome this issue. In fact, Abawi et al. [32] found a reduction in HRQoL during the COVID-19 pandemic according to parents’ report in the same generic measure, PedsQL, used in the current study.

Regarding the HRQoL sub-domains, our results indicated complex differences in adolescents’ HRQoL during both time periods: higher physical HRQoL alongside lower school HRQoL during the COVID-19 outbreak, compared to their peers’ reports before the pandemic. Adolescents in the COVID-19 group reported during Israel’s first lockdown, which included restricted access to public spaces and entertainment, closed high schools, and educational activities conducted online. The change in the learning environment may have adversely affected their school-related HRQoL. These results are consistent with other studies reporting that adolescents’ HRQoL during the pandemic was affected by the changing routine and educational setting, alongside deterioration of relationships with peers, socioeconomic status, and migration background [33,35].

Surprisingly, the physical HRQoL was higher in the pandemic group than in the pre-pandemic group. These results can be explained by the items included in this part of the PedsQL, which describe difficulties in running, physical activities, fatigue, etc. During the COVID-19 lockdown, there was reduced opportunity to participate in physical activity. Therefore, adolescents participated less in physical activities [36], thus reporting fewer difficulties and pain manifestations in this area than before COVID-19 lockdowns. The low participation in physical activity may result in a lack of encounter with difficulty, resulting in a report of good physical HRQoL. Another explanation is the effect on adolescents of extended sleep and rest time typical during lockdowns [9], which may have resulted in reduced fatigue.

Our results regarding resilience factors were preliminary, since there were no previous studies assessing adolescents’ SOC and hope during the pandemic. This study found that during the COVID-19 outbreak, adolescents reported higher SOC, compared to their peers before the outbreak, but no differences in hope levels. In the salutogenic model, SOC is a major resource for maintaining health when facing dangers and challenges [15,19]. A person’s SOC develops through experiences in their life, which allows them to anticipate logical and predictable conduct in the world. The SOC is expressed in efficient and repeated use of coping resources at their disposal [17]. Accordingly, SOC is a dynamic construct that varies in adolescence and becomes stable only by age 30 [23]. Because SOC is still changing during adolescence and is open to enhancement [27], it is important to help children in this developmental stage to process their experiences of successfully coping with events and challenges in order to empower and support this factor.

The importance of SOC to health was also found in its correlation with physical HRQoL, which was significantly stronger in the group during COVID-19, compared to the pre-outbreak group. One possible explanation may relate to the perception of the pandemic as consistent. The level of coherence is determined, among other things, by the extent to which external events are experienced as consistent [17]. In the COVID-19 lockdown, authorities around the world put coordinated effort into advocacy and presenting information on prevention, the purpose of the restrictions, and the importance of compliance. These clarifications may have contributed to the comprehensibility component of SOC among adolescents, perhaps even improving it during the lockdown.

In contrast to SOC, no significant difference was found between pre-lockdown and lockdown periods in levels of hope. Hope involves a person’s beliefs about their abilities to achieve their aims in varied situations [18], regardless of health conditions. Hence, this perceived ability would not change because of the COVID-19 challenge. Moreover, hope has two dimensions: dispositional hope and state hope. The first is stable across situations and times, while the second reflects particular settings and is measured in a given moment [20]. Our study measured dispositional hope, thus it is not surprising that no changes were found.

Finally, our findings reinforce the role of SOC and hope as main variables in HRQoL, emphasizing their importance as resilience factors in general and during the COVID-19 lockdown in particular. In both periods, the correlations between overall HRQoL and resilience factors were found to be high. Moreover, the results revealed that in the presence of SOC and hope, socio-demographic characteristics and the pandemic had no predictive value regarding HRQoL. Therefore, these resilience factors might serve as buffers for unfavorable changes in socio-demographic characteristics and other stressors in adolescents. Our findings are similar to previous studies among adults during the COVID-19 pandemic, which found not only that SOC and hope were correlated with mental-health components [30,37] and predicted psychological distress [38] and life satisfaction [39], but that they also moderated the link between COVID-19 diagnosis and psychological well-being [40].

### Limitations

Several limitations are evident in this study. First, the study included a fairly small convenience sample from high socio-demographic settings (e.g., highly educated parents and high rates of married parents, compared to the general population). Therefore, the results may have been influenced by the participants’ cultural environment and lifestyle [41]. Second, the two groups were composed of different adolescents; in light of this, the results of the research should be taken with caution. Furthermore, it is reasonable to believe that confounding variables, which were not sampled (for example, changes in family routines during the lockdown) could have had a significant effect on the different HRQoL domains and that the relationship between resilience factors and HRQoL might be better explained by their shared associations with such an untested variable. Third, the data covered only the beginning of the pandemic. Additional research should assess the prolonged effects of repeated lockdowns, social distancing, and other pandemic characteristics on HRQoL and resilience factors. Finally, in this study, the HRQoL was tested using a generic measure and not specifically to the COVID-19 pandemic. Furthermore, the PedsQL needs additional investigation about his specificity or sensitivity in his Hebrew version [26]. 

## 5. Conclusions

This study indicated lower HRQoL among healthy adolescents during the outbreak, especially lower school-related HRQoL; a higher SOC, compared to before the outbreak; and no difference in hope levels between the study groups. Increased SOC and hope were associated with increased overall HRQoL levels. These results contribute to the recent COVID-19-inspired trend of exploring resources that enable individuals to deal with stress and adapt to changing circumstances [12,15,16,24,30,38]. According to our findings, hope and SOC play an important role in determining an adolescent’s health outcomes. Given the priority of supporting an individual during development, it makes sense to raise awareness of these resilience factors and to address them in encounters and interventions among adolescents in educational and therapeutic settings. By doing so, adolescents’ resilience can be strengthened, and their health maintained in challenging situations such as the COVID-19 pandemic.

## Figures and Tables

**Figure 1 ijerph-19-03157-f001:**
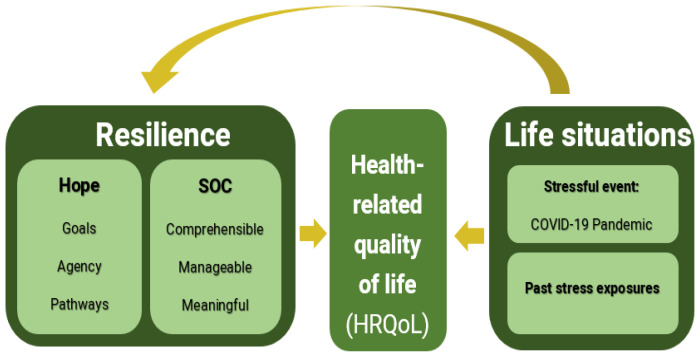
The study model: The reciprocal relationships between resilience factors, HRQoL and Life situations (e.g., COVID-19).

**Table 1 ijerph-19-03157-t001:** Group differences in demographic characteristics.

	Before	During	t/χ^2^
	COVID-19	COVID-19	
	Outbreak	Outbreak	
	*n* = 84	*n* = 64	
Gender N (%)			0.086 ^a^
Boys	41 (48.8)	32 (50)	
Girls	43 (51.2)	32 (50)	
Age M (SD)	15.15 (1.59)	14.77 (1.93)	−1.305
Mother’s education N (%)			19.154 ***
High school	23 (27.38)	1 (1.6)	
Professional studies	10 (11.9)	6 (9.4)	
College degree	51 (60.7)	57 (81.9)	
Father’s education N (%)			15.017 **
High school	23 (27.38)	5 (7.82)	
Professional studies	5 (5.95)	8 (12.5)	
College degree	56 (66.67)	51 (79.69)	
Parents’ marital status N (%)			5.526
Married	81 (96.43)	60 (93.75)	

M—mean; SD—standard deviation; ^a^ Fisher’s Exact Test; ** *p* < 0.01; *** *p* < 0.001.

**Table 2 ijerph-19-03157-t002:** Group differences according to PedsQL.

	Before	During	F (1146)	η²
	COVID-19	COVID-19		
	Outbreak	Outbreak		
	*n* = 84	*n* = 64		
	M (SD)	M (SD)		
Physical HRQoL	72.12 (19.94)	78.07 (14.24)	3.710 *	0.027
Psychological HRQoL	76.19 (11.85)	73.81 (12.66)	2.124	0.015
Emotional HRQoL	71.43 (16.91)	69.53 (17.27)	1.761	0.003
Social HRQoL	85.48 (13.82)	86.23 (14.57)	0.000	0.001
HRQoL in school	71.14 (16.46)	65.66 (15.77)	5.748 *	0.028
Total HRQoL	76.01 (11.3)	74.58 (13.14)	0.371	0.003

HRQoL-Health-related quality of life; PedsQL-Pediatric Quality of Life Inventory; * *p* < 0.05.

**Table 3 ijerph-19-03157-t003:** Pearson correlations between PedsQL and SOC and Hope Scales.

	SOC		Hope	
	Before	During	Before	During
	COVID-19	COVID-19	COVID-19	COVID-19
	Outbreak	Outbreak	Outbreak	Outbreak
Physical HRQoL	0.17	0.57 **	0.24 *	0.45 ***
Psychological HRQoL	0.48 ***	0.57 ***	0.54 ***	0.61 ***
Total HRQoL	0.49 ***	0.62 ***	0.52 ***	0.60 ***

SOC-sense of coherence; * *p* <0.05; ** *p* < 0.01; *** *p* < 0.001.

**Table 4 ijerph-19-03157-t004:** Regression analysis for predictors of total PedsQL score.

	B	SE B	β
Age	0.45	0.492	0.064
Gender	1.267	1.731	0.052
Mother’s education	−1.300	1.366	−0.076
Father’s education	−0.007	0.807	−0.001
Group/Time	1.946	1.852	0.08
SOC	6.078	1.359	0.374 ***
Hope	4.877	1.266	0.320 ***

SOC-sense of coherence; *** *p* < 0.001.

## Data Availability

The data sets generated during and/or analyzed during the current study are not publicly available due to patients’ privacy but are available from the corresponding author on reasonable request.

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
