# Peer review of "Adolescents and Resilience: Factors Contributing to Health-Related Quality of Life during the COVID-19 Pandemic"

_ijerph, 2022, doi:10.3390/ijerph19063157_

Round 1

Reviewer 1 Report

Thank you for the opportunity to see this interesting work. Authors should make corrections at once.
1. in section 2.2, the Pediatric Quality of Life Inventory Version 4.0 (PedsQL) (Generic Measures or Specific Measures) should be included in the form category.
2. If there is a Generic Measures form, the lack of differences in QoL from before and during the pandemic may be due to the low sensitivity and specificity of the form. This fact should be discussed in the discussion and constraints of the work. Future research should be done using the Specific Measures form.
3. In the discussion, the authors should also refer to 

1. Fenny DH , Furlong W i wsp. A framework for assessing Heath –related quality of life among children with cancer. Int J Cancer 1991; suppl. 12 2-9 2. Fokens WJ, Lund VJ, Bachert C et al. European position paper on rhinosinusitis and nasal polyps. Rhinology 2020, suppl 20:1-134 3. Kay DJ, Rosenfeld RM. Quality of Life for Children with persistent sinonasal symptoms. Otolaryngol. Head Neck Surg. 2003; 128, 17-26 4. Niedzielski A, Chmielik LP, Kasprzyk A, Stankiewicz T, Mielnik-Niedzielska G. Health-Related Quality of Life Assessed in Children with Adenoid Hypertrophy. Int J Environ Res Public Health. 2021 Aug 25;18(17):8935. doi: 10.3390/ijerph18178935. PMID: 34501525;

Author Response

Dear Editor,

Thank you for the opportunity to revise our manuscript entitled" Adolescents and resilience: Factors contributing to health-related quality of life during the COVID-19 pandemic"

All reviewers' comments have been addressed, and a revised manuscript with tracked changes has been submitted. Following is a list of revisions according to the reviewer’s comments.

Reviewer 1#
1. in section 2.2, the Pediatric Quality of Life Inventory Version 4.0 (PedsQL) (Generic Measures or Specific Measures) should be included in the form category.

Thank you for this important note. The PedsQL is a generic measure for HRQoL among children and adolescents, we added reference to the type of measure in section 2.2 (Lines 153, 157) and added this important point to the limitation section (Lines 400-403).
2. If there is a Generic Measures form, the lack of differences in QoL from before and during the pandemic may be due to the low sensitivity and specificity of the form. This fact should be discussed in the discussion and constraints of the work. Future research should be done using the Specific Measures form.

Due to your comments, we have added a paragraph to the discussion that deals with this issue (Lines 290-308)

  1. In the discussion, the authors should also refer to:

Fenny DH , Furlong W i wsp. A framework for assessing Heath –related quality of life among children with cancer. Int J Cancer 1991; suppl. 12 2-9 2.

Fokens WJ, Lund VJ, Bachert C et al. European position paper on rhinosinusitis and nasal polyps. Rhinology 2020, suppl 20:1-134 3.

Kay DJ, Rosenfeld RM. Quality of Life for Children with persistent sinonasal symptoms. Otolaryngol. Head Neck Surg. 2003; 128, 17-26 4.

Niedzielski A, Chmielik LP, Kasprzyk A, Stankiewicz T, Mielnik-Niedzielska G. Health-Related Quality of Life Assessed in Children with Adenoid Hypertrophy. Int J Environ Res Public Health. 2021 Aug 25;18(17):8935. doi: 10.3390/ijerph18178935. PMID: 34501525;

Thank you for your offer. The first and last referrals have been added to the discussion of the taxonomy and theoretical ideas offered in some of these articles (Lines 290-308).

Reviewer 2 Report

Hello,

Thanks for letting me review your work: I have a couple of comments for the manuscript:

  1. What does Mage stand for?
  2. How were the adolescents recruited? You said they were recruited differently.
  3. The latest coronavirus (COVID-19) is an infectious disease that should be restated into The latest coronavirus leads to or causes an infectious disease.
  4. What is the salutogenic model - do you have a graph of the study model?
  5. Was there a power analysis that drove the selection of the subject number?
  6. Where was this study conducted?
  7. What are the values for the Children's Hope scale in internal consistency and test-retest reliability?

Author Response

Dear Editor,

Thank you for the opportunity to revise our manuscript entitled" Adolescents and resilience: Factors contributing to health-related quality of life during the COVID-19 pandemic"

All reviewers' comments have been addressed, and a revised manuscript with tracked changes has been submitted. Following is a list of revisions according to the reviewer’s comments.

Reviewer 2#

1.What does Mage stand for?

Mage in an abbreviation for mean age. We change the term in the text in section 2.1 (Lines 122 + 124).

2.How were the adolescents recruited? You said they were recruited differently.

Both groups recruited in the same manner. We added in section 2.3 (procedure section) clarifies how the data was collected (Lines 185-190).

3.The latest coronavirus (COVID-19) is an infectious disease that should be restated into The latest coronavirus leads to or causes an infectious disease.

We added this good comment to the introduction section (Lines 32-34).

4.What is the salutogenic model - do you have a graph of the study model?

We added further explanation about the salutogenic model (Lines 82-95) and added a graph of the study model (Lines 114-118).

5.Was there a power analysis that drove the selection of the subject number?

We did conducted a power analysis. We added reference in the Participants section (lines 124-128).

6.Where was this study conducted?

This study was conducted in Israel. We added explicit reference for the place where the study took place in sections 2.1 (participants) and 2.3 (procedure). 

7.What are the values for the Children's Hope scale in internal consistency and test-retest reliability?

We added the values in the measure descriptions (Lines 177-178).

Reviewer 3 Report

Thanks for recommending me as a reviewer. This study aimed to examine health-related quality of life of adolescents before and during the COVID-19 outbreak, and its relationship to resilience embodied in hope and a sense of coherence. If authors complete minor revisions, the quality of the study will improve.

  1. The introduction section is well written. Lines 97-113 are more concisely abbreviated, and the results of previous studies are recommended to be moved to the discussion section.

2. In the footnotes of Table 1, it is recommended to unify the significance level to *<0.05. Otherwise, **, *** as well as * are required in footnotes.

3. line 197: "The significance level for obtaining the research hypothesis was p < .05." - In this study, significance levels of 0.01 and 0.001 are also indicated. If this sentence is described, it is recommended to unify the notation for the significance level to <0.05.

Author Response

Dear Editor,

Thank you for the opportunity to revise our manuscript entitled" Adolescents and resilience: Factors contributing to health-related quality of life during the COVID-19 pandemic"

All reviewers' comments have been addressed, and a revised manuscript with tracked changes has been submitted. Following is a list of revisions according to the reviewer’s comments.

Reviewer 3#

1.The introduction section is well written. Lines 97-113 are more concisely abbreviated, and the results of previous studies are recommended to be moved to the discussion section.

We changed the introduction by erasing lines 101 to 113. We rewrite these sentences at the beginning of the discussion (lines 249-258). Additionally, we removed some of the sentences since similar information appears in the discussion in lines 374-381.

  1. In the footnotes of Table 1, it is recommended to unify the significance level to *

<0.05. Otherwise, **, *** as well as * are required in footnotes.

Thank you, we added the 0.05 level of significance for Table 1 notes.

  1. line 197: "The significance level for obtaining the research hypothesis was p < .05." - In this study, significance levels of 0.01 and 0.001 are also indicated. If this sentence is described, it is recommended to unify the notation for the significance level to <0.05.

Thank you, it is an important comment. However, the level of significance included p < 0.05 - in line 215; table 2; table 3. This was the reason the p value set at 0.05. However, we did unify the notations in all of the tables.

Round 2

Reviewer 1 Report

The work is interesting and provides a good basis for further work